# Genomic Characterization of IMP-Producing *Pseudomonas aeruginosa* in Bulgaria Reveals the Emergence of IMP-100, a Novel Plasmid-Mediated Variant Coexisting with a Chromosomal VIM-4 [note 1]

**DOI:** 10.3390/microorganisms11092270

**Published:** 2023-09-09

**Authors:** Ivan Stoikov, Ivan N. Ivanov, Deyan Donchev, Deana Teneva, Elina Dobreva, Rumyana Hristova, Stefana Sabtcheva

**Affiliations:** 1National Reference Laboratory for Control and Monitoring of Antimicrobial Resistance, Department of Microbiology, National Center of Infectious and Parasitic Diseases, 26 Yanko Sakazov Blvd., 1504 Sofia, Bulgaria; ivanoov@gmail.com (I.N.I.); deyandonchev@ncipd.org (D.D.); teneva.deana@gmail.com (D.T.); elina_g@abv.bg (E.D.); rumyana.hristova@googlemail.com (R.H.); 2Laboratory for Clinical Microbiology, National Oncology Center, 6 Plovdivsko pole Str., 1797 Sofia, Bulgaria; stefanasabtcheva@gmail.com

**Keywords:** *Pseudomonas*, metallo-beta-lactamase, blaIMP, novel allele, plasmid

## Abstract

Multidrug-resistant (MDR) *Pseudomonas aeruginosa* infections represent a major public health concern and require comprehensive understanding of their genetic makeup. This study investigated the first occurrence of imipenemase (IMP)-carrying *P. aeruginosa* strains from Bulgaria. Whole genome sequencing identified a novel plasmid-mediated IMP-100 allele located in a a novel *In4886* integron embedded in a putative *Tn7700* transposon. Two other closely related chromosomal IMP variants, IMP-13 and IMP-84, were also detected. The IMP-producers were resistant to last-line drugs including cefiderocol (CFDC) (two out of three) and susceptible to colistin. The IMP-13/84 cassettes were situated in a *In320* integron inserted in a *Tn5051*-like transposon as previously reported. Lastly, the p4782-IMP plasmid rendered the PA01 transformant resistant to CFDC, suggesting a transferable CFDC resistance. A variety of virulence factors associated with adhesion, antiphagocytosis, iron uptake, and quorum sensing, as well as secretion systems, toxins, and proteases, were confirmed, suggesting significant pathogenic potential consistent with the observed strong biofilm formation. The emergence of IMP-producing MDR *P. aeruginosa* is alarming as it remains unsusceptible even to last-generation drugs like CFDC. Newly detected IMP-100 was even located in a CFDC-resistant XDR strain.

## 1. Introduction

*Pseudomonas aeruginosa* is a member of the ESKAPE (*Enterococcus faecium*, *Staphylococcus aureus*, *Klebsiella pneumoniae*, *Acinetobacter baumannii*, *Pseudomonas aeruginosa*, and *Enterobacter species*) group of highly resistant pathogens and represents a formidable challenge in healthcare settings [1]. In the past decade, the emergence of carbapenem resistance among *P. aeruginosa* isolates has become a global concern as carbapenems are among the last options considered for treatment [2,3]. Carbapenem resistance is commonly associated with mutations in the outer-membrane porin OprD, overexpression of efflux pumps, and horizontal acquisition of carbapenemase genes, including both serine- and metallo-β-lactamases (MBLs) [3,4,5,6].

Beta-lactamases encoded by *bla* genes are grouped into four classes (A, B, C, D) according to the widely used Ambler classification. Carbapenemase activity in clinical strains of *P. aeruginosa* due to *Klebsiella pneumoniae* carbapenemase (KPC) and certain Guiana extended-spectrum β-lactamase (GES) variants (Ambler class A) are less frequently reported, similar to the oxacillinases (OXA) from Ambler class D with activity against carbapenems [7]. MBLs (Ambler class B) are characterized by their dependency on one or two zinc cations for enzymatic activity [4]. Various types of acquired MBLs have been identified in *P. aeruginosa*, including imipenemase (IMP), Verona integron metallo-beta-lactamase (VIM), São Paulo metallo-beta-lactamase (SPM), Germany imipenemase (GIM), New Delhi MBL (NDM), Florence imipenemase (FIM), and recently Belém imipenemase (BIM) [7,8]. Among these, IMP- and VIM-type enzymes are the most abundant and are of particular clinical importance, as they efficiently inactivate most β-lactam antibiotics, with the exception of monobactams [9,10]. The first identification of IMP-1 metallo-β-lactamase was reported in 1988 in a *P. aeruginosa* strain isolated from Japan [11]. Since then, the emergence and spread of IMP-carrying *P. aeruginosa* has been reported globally, often associated with international high-risk clones such as ST111, ST233, and ST235 [12]. The IMP-type enzymes represent a highly heterogeneous group forming at least seven phylogenetic clusters and frequently found as gene cassettes in class 1 integrons [4,13,14].

Integrons play a crucial role in capturing and disseminating antibiotic resistance genes including MBLs [4] as they are often associated with large transposon structures found on plasmids or chromosomes [13,15]. Transposable elements utilize specific transposase-mediated mechanisms for their insertion and excision within the bacterial genome. Composite transposons are flanked by insertion sequence (IS) elements. Unit transposons, on the other hand, encode an excision/integration-associated enzyme, recombinase or resolvase, along with accessory genes such as resistance genes, within a single genetic unit. Conjugative transposons, also referred to as integrative conjugative elements (ICEs), carry genes for excision, conjugative transfer, and integration, often accommodating a diverse repertoire of accessory genes, including antibiotic resistance genes [16]. Resistance plasmids could be directly studied from genomic data by either replicon or MOB-typing schemes based on the diversity of replicative and relaxase proteins, respectively. MOB-typing is more sensitive and amenable to plasmids in *Pseudomonas* spp. [17].

*P. aeruginosa* isolates in Bulgaria display a range of carbapenemases, of which VIM-type enzymes are commonly reported [18,19]. The co-occurrence of NDM-1 and GES-5 carbapenemases was recently identified [20]. The detection of OXA-50 carbapenemase has also been documented, although currently it is being considered intrinsic in *P. aeruginosa* [21].

In this study, we performed genomic analysis of three clinical *P. aeruginosa* isolates, revealing to our knowledge the first occurrence of IMP-carrying *P. aeruginosa* in Bulgaria. Furthermore, among the isolates, there was one that harbored a novel variant of the IMP carbapenemase, officially denoted as IMP-100 upon its submission in the NCBI GenBank database. This variant was situated on a multidrug-resistant plasmid which we termed p4782-IMP and coexisted with a chromosomally encoded blaVIM-4 gene.

## 2. Materials and Methods

### 2.1. Strains

The three strains (Paer3541, Paer3796A, and Paer4782MK) were initially isolated in Sofia, Bulgaria, between 2018 and 2022. Paer3541 was obtained from the throat swab of a hospitalized patient, Paer3796A was derived from the urine sample of an individual in an outpatient setting, and Paer4782MK was isolated from the blood culture of a patient with acute myeloid leukemia. Strains were sent to the National Reference Laboratory, Department of Microbiology, within the National Center of Infectious and Parasitic Diseases, Sofia, Bulgaria, for carbapenem resistance confirmation purposes (Appendix A). These three strains were the only IMP-type carbapenemase-carrying *P. aeruginosa* strains to date.

### 2.2. Phenotypic and Molecular Analysis

Strains were cultured on Columbia agar (Diachim AD, Sofia, Bulgaria) at 35 °C overnight. A single colony of each strain was identified via MALDI Biotyper (Bruker Daltonics GmbH & Co. KG, Bremen, Germany) with MALDI Reference 2022 Library v4.0.0. Antimicrobial Susceptibility Testing (AST) was performed using MICRONAUT-S Pseudomonas MIC and UMIC^®^ Cefiderocol assay (Bruker Daltonics GmbH & Co. KG, Bremen, Germany). The interpretation of the AST results was in accordance with EUCAST clinical breakpoints v13.0.

Initially, the isolates were analyzed for carbapenemase activity by a modified CarbaNP test [22]. In parallel, an in-house carbapenemase gene detection multiplex PCR was performed with the PCR components, conditions, and protocols described in detail in Appendix A. Primer pairs for each gene were obtained from previously published sources, including SIM, SPM, OXA-48-like, GES, and KPC [23,24,25,26] along with additional primer pairs for IMP, VIM, and NDM [27]. QIAxcel Advanced high-resolution capillary electrophoresis system was used following the PCR (Qiagen, Hilden, Germany) with protocol 0M800 (3 kV for 800 s) for precise size estimation.

The antimicrobial resistance gene expression assay of AMR-associated genes *mexA*, *mexC*, *mexE*, *mexX*, *ampC*, and *oprD* was performed as previously described [5,28], and results interpreted according to [29]. Biofilm formation was quantified by the modified crystal violet (CV) assay [30].

### 2.3. Genomic and Plasmid DNA Extraction and Whole Genome Sequencing (WGS)

Total genomic DNA extraction was performed with PureLink™ Genomic DNA Mini Kit (Thermo Fisher Scientific, Missouri, TX, USA) according to the manufacturer’s instructions except that all homogenization steps were carried out by pipetting. Plasmid DNA was acquired using NucleoSpin Plasmid Mini kit for plasmid DNA (Macherey-Nagel, Düren, Germany) following the “low-copy plasmid” protocol. Short-read next-generation sequencing (NGS) was performed using Illumina DNA Prep kit for sequencing libraries preparation and MiSeq V3 (2 × 300 bp) for strains Paer3541 and Paer3796A, whereas Paer4782MK was sequenced on NextSeq 550 with V2.5 (2 × 150 bp) mid output flow cell (Illumina, Inc., San Diego, CA, USA). The same DNA extract without additional size-selection was used for long-read sequencing on a MinION Mk1C with the Rapid Barcoding Kit 96 (SQK-RBK110.96) and FLO-MIN106D (R9.4.1) (Oxford Nanopore Technologies, Oxford, UK). The final purification step of the library pool was performed with 0.4x SPRI magnetic particles as recently suggested for removal of DNA fragments <1.5 kb [31].

### 2.4. Cloning, Transformation, and Conjugation/Mating Experiments

An 1150 bp fragment amplified with primers IMP-new_Clon_F and R comprising the complete IMP-100 (OR004774) open reading frame (ORF), its promotor region, and part of the *IntI* gene was cloned into the pET28a T7pCONS TIR-2 sfGFP vector (cat. No 154464, Addgene, Inc., Watertown, MA, USA) using the FastCloning protocol [32]. The insert amplification and vector linearization PCR are described in detail in Appendix A. A schematic representation of the cloning design is available in Appendix A. The PCR products mixed at 1:1 ratio were digested with *Dpn*I for 2 h at 37 °C.

Plasmid DNA from the Paer4782MK strain was transformed into *P. aeruginosa* PA01^RifR^. The transformation protocols for both the cloned IMP-100 and the whole p4782-IMP plasmid preparation were identical [33]. Briefly, *E. coli* NEB-10 (New England Biolabs, Inc., Ipswich, MA, USA) or *P. aeruginosa* PA01^RifR^ recipients were incubated on Columbia agar at 35 °C overnight. A few colonies were inoculated into 10 mL BHI broth and agitated at 200 rpm at 35 °C until log phase (0.6 ± 0.05 OD) was achieved. Then, 1.4 mL suspension was aliquoted into 1.5 mL tubes and centrifuged at 8000× *g* for 2 min and the supernatant was discarded. Cell pellets were washed with 1 mL RT molecular-grade water twice by gentle resuspension. Cell pellets were resuspended in 80 µL of RT molecular-grade water. The digested vector and insert mix (20 µL) or plasmid DNA were added, gently mixed, and immediately transferred into 2 mm gap electroporation cuvettes (Avantor, Inc., Radnor, PA, USA). The electroporation was performed at 2500 mV on an Eporator^®^ (Eppendorf, Hamburg, Germany). Immediately, 1 mL of BHI broth (RT) was added, gently mixed, and incubated at 35 °C for 2 h. Fresh BHI agar plates containing 50 mg/L kanamycin and 8 mg/L ceftazidime were streaked with 50 µL of the transformants.

The spontaneous transferability of plasmids carrying resistance determinants was investigated by filter mating [34] and the combined mating technique [35]. In an attempt to increase the conjugation efficiency in *Pseudomonas*, additional steps were implemented in the protocol as previously described [36]. Rifampicin-resistant *E. coli* C600 and a spontaneous mutant of *P. aeruginosa* PA01^RifR^ were used as recipient strains. Transconjugants were selectively cultured on MacConkey agar supplemented with ceftazidime (30 mg/L) and rifampicin (200 mg/L) (Sigma Chemical Co., St Louis, MO, USA).

### 2.5. Bioinformatic Analysis

A quality check of both long and short reads was performed with FastQC v0.11.9 (https://www.bioinformatics.babraham.ac.uk/projects/fastqc, accessed on 8 June 2023). Quality trimming and filtering of the raw reads was conducted with fastp v0.23.2 [37] and filtlong v0.2.1 (https://github.com/rrwick/Filtlong, accessed on 8 June 2023) for short and long reads, respectively. First, long-read-only assemblies were produced with Flye v0.2.1 [38] and when no circular genome was achieved, hybrid assembly was attempted with Unicycler v0.4.8 [39]. The resulting assemblies were subsequently polished with tools such as Polypolish v0.5.0 [40], POLCA [41] and MEDAKA v1.7.3 (ONT, https://github.com/nanoporetech/medaka, accessed on 15 June 2023). Assembly quality was assessed by multiple tools such as Quast v 5.0.2 [42], BUSCO v5.4.6 [43], and CheckM v1.2.1 [44]. After genome quality evaluation, strain identification was performed with rMLST [45] and KmerFinder v 3.0.2 [46] with a database version from 11 July 2022. Then, genomes were annotated with Bakta v 1.7 [47] with database v5.0-full. Antibacterial resistance, phenotypic prediction, and virulence determinants were identified with AMRFinderPlus v3.11.4 [48] and ResFinder v4.3.1 [49] with database version 2022-05-24 and VFanalyzer (database version from 5 January 2023) [50], respectively. Mob-suite v 3.1.4 [17] was used for plasmid analysis. Analysis of ICEs was carried out with ICEfinder [51]. MLST profiles were inferred from sequence data using mlst v2.23.0 (Seemann T, mlst Github https://github.com/tseemann/mlst, accessed on 8 June 2023). In silico serotyping was performed using the tool PAst 1.0 distinguishing 11 serogroups by BLAST analysis of the O-specific antigen cluster. [52]. The functionality of the outer-membrane porin *oprD* was analyzed in silico using the recently described PorinPredict tool [53].

### 2.6. Data Availability

The complete genomes of the three strains were deposited in the European Nucleotide Archive (ENA) under project accession PRJEB62425. The genomes were assigned the following accession numbers: Paer4782MK—ERZ18545754, Paer3541—ERZ18545673, Paer3796A—ERZ18545648. In the case of Paer4782MK, a novel variant of IMP-100, a subclass B1 metallo-beta-lactamase, was detected, validated, and submitted to GenBank with accession number OR004774. The putative novel transposon was registered in The Transposon Registry (https://transposon.lstmed.ac.uk/, accessed on 28 August 2023) under *Tn7700*.

## 3. Results

### 3.1. AMR and Biofilm Formation

Antimicrobial susceptibility testing (AST) showed diverse resistance patterns among the tested strains. Paer4782MK exhibited a high level of resistance to multiple classes of antibiotics, including all beta-lactams except aztreonam, aminoglycosides, and fluoroquinolones, consistent with an XDR phenotype. Paer3541 demonstrated resistance to cephalosporins and fluroquinolones but remained susceptible to carbapenems and aminoglycosides (except tobramycin). Paer3796A displayed resistance to all tested cephalosporins, carbapenems (except imipenem/relebactam), fluoroquinolones (except levofloxacin), and aminoglycosides (except amikacin). However, all three strains remained susceptible to colistin despite resistance to novel cephalosporin/inhibitor combinations like ceftazidime/avibactam and ceftolozane/tazobactam. Notably, both the Paer4782MK and Paer3796A strains exhibited resistance to CFDC, a novel siderophore cephalosporin only recently approved for treatment of infections from MDR Gram-negatives [54]. The AST profiles of the three *P. aeruginosa* isolates are available in Table 1.

The modified CarbaNP test confirmed the presence of carbapenemases in all three isolates. Initially, the PCR results for IMP genes were negative. However, by employing alternative PCR primers [27], we successfully detected the presence of *bla*_IMP_ in all three isolates and *bla*_VIM_ in Paer4782MK.

We observed a positive correlation between the notable upregulation of the *mexXY* multidrug efflux operon in Paer3541 and the elevated MICs for levofloxacin and ciprofloxacin, and also between low to moderate *mexXY* expression and lower fluoroquinolone MIC values in Paer3796A. Despite Paer4782MK being characterized by the moderate expression of the *mexCD*, its association with the resistance profile remained uncertain as the strain already carried resistance genes for the majority of *mexCD* targets as seen from the WGS results. Interestingly, all three strains exhibited negative expression of the outer membrane porin *oprD*. The gene expression levels are available in Appendix A. Lastly, all strains exhibited strong biofilm production.

### 3.2. Genome Quality Assessment and Features

All three genomes achieved a completeness score close to 100%, indicating that the majority of the expected conserved genes were present. A high level of genome integrity without duplications was evidenced by the 99.2% single-copy genes detected. No fragmented or missing conserved genes were identified. In line with these findings, CheckM revealed relatively low foreign DNA contamination levels ranging from 0.63% to 0.7%. Detailed information on assembly statistics and genome quality assessment is available in Appendix A.

Two plasmids, p4782-IMP (61.5 Kbp) and p4782_002 (290.8 Kbp), were identified in Paer4782MK. The plasmid p4782-IMP (OX638703.1) was related to the MOB_F_ and MPF_T_ plasmid pMOS94-like family. Notably, this plasmid carried the novel *bla*_IMP-100_ (OR004774) allele, in addition to other genes associated with AMR. The plasmid p4782_002 (OX638702.1) was untypable and lacked both resistance and virulence determinants; thus it was excluded from further analysis. The two plasmids detected in Paer3541 were p3541_1 (179.3 Kbp, OX638611.1) and p3541_2 (41.5 Kbp, OX638612.1). Similar to p4782-IMP, the plasmid p3541_2 was MOB_F_ and MPF_T_ and was found to harbor a single *aac(6′)-29* gene, encoding an aminoglycoside acetyltransferase. The plasmid p3541_1 had unknown MOB and MPF types, similar to p4782_002. For Paer3796A, a single plasmid p3796A (178.5 Kbp, OX638565.1) with unknown MOB and MPF types was identified and did not harbor any resistance or virulence determinants. Additional information regarding plasmid analysis is available in Appendix A.

Paer3541 and Paer3796A were assigned to ST621, a clone previously associated with IMP carriage [55], while Paer4782MK belonged to the international high-risk clone ST233. In silico serotyping revealed that Paer3541 and Paer3796A were serogroup O4, while Paer4782MK was O11.

### 3.3. Detection of AMR Determinants

The AMR screening uncovered a diverse array of genes (Figure 1). All three isolates demonstrated the presence of an IMP carbapenemase, with the unique IMP-100 allele identified in a pMOS94-like plasmid in Paer4782MK. Additionally, strain Paer4782MK also harbored *bla*_VIM-4_. Two other closely related chromosomal IMP variants, IMP-13 and IMP-84, were detected in Paer3541 and Paer3796A, respectively. Additional acquired extended-spectrum beta-lactamases (ESBL) such as PER-1 and multiple genetic determinants associated with resistance to most non-beta-lactam antimicrobial groups were present as well.

In silico analysis of the *oprD* gene revealed that both Paer3541 and Paer3796A harbor an intact and functional *oprD* porin. Despite the identification of a missense mutation (S325F) in the *oprD* gene of Paer3796A, the gene appeared intact. Conversely, a truncated *oprD* protein was evident in Paer4782MK, potentially compromising its function (Appendix A). No known mutations related to efflux overexpression were detected.

### 3.4. Detection of Virulence Determinants

All three isolates exhibited virulence factors related to adherence and motility, including flagella, type IV pili biosynthesis, and twitching motility. Alginate biosynthesis genes known for promoting bacterial resilience against harsh conditions and host immune responses [56] were detected in all isolates. However, certain antimicrobial-related factors (*phzC2, phzD2*, *phzE2*, *phzF2*, and *phzG2*) were absent from all strains compared to the reference strain PA01. Phospholipases C and the hemolytic *plcH* [57] were identified in all strains, while phospholidase D was absent from Paer4782MK. Complete siderophore biosynthesis operons were present in all strains, with the exception of the pyoverdine production gene *pvdD,* which was absent from Paer4782MK, indicating impaired pyoverdine synthesis [58]. Additional virulence factors, including alkaline protease (AprA), elastase (LasA and LasB), protease IV (PrpL), and exotoxin A (ToxA), as well as quorum-sensing components such as acylhomoserine lactone synthase (HdtS), transcription factors (LasI, RhlI), and receptors (LasR, RhlR), as well as the GacS/GacA two-component system and type VI secretion system (H-T6SS) components, were found in all three isolates. The rhamnolipid biosynthesis operon *rhl* associated with host cell infiltration and biofilm formation was also present in the studied strains [59,60]. T3SS effectors, including *exoS*, *exoT*, and *exoY*, involved in manipulating host cell signaling pathways and immune responses, along with T3SS genes such as *pcrV*, *popB*, *popD*, and *pscF*, were common across all our strains indicating a functional T3SS machinery capable of delivering effector molecules into host cells. All isolates harbored the hydrogen cyanide operon (HcnABC), a potent toxic compound known to disrupt cellular processes and impact aerobic respiration [61]. Furthermore, *katG*, involved in oxidative stress protection [62], was found exclusively on the p4782-IMP in Paer4782MK. A summary of the virulence screening results can be found in Appendix A.

### 3.5. Plasmid Analysis

The complete sequence of plasmid p4782-IMP was subjected to a BLAST search against the nr/nt database of NCBI, and a neighbor-joining tree was constructed with the most similar plasmids (n = 19). Interestingly, plasmid p3541_2 was identified as closely related to p4782-IMP. Additionally, all plasmids were screened for AMR genes and the results were illustrated as a heatmap and linked to the phylogenetic tree (Figure 2).

Upon observation, cluster C1 was found to encompass three plasmids with unknown MOB type that encoded KPC enzymes. The remaining clusters C2–C4 comprised plasmids from the MOB_F_ and MPF_T_ families. C2 was similar to C3 plasmids, but unlike them carried only the KPC enzyme. C3 and C4 both exhibited relatedness to the pMOS94-like plasmids [63]. Interestingly, C3 plasmids had lower density of AMR-associated genes compared to C4. Despite the high abundance of AMR genes found on plasmid p4782-IMP similar to C4 plasmids, it was grouped within the lower-resistance cluster C3 due to the larger degree of similarity of the backbone genes to C3.

Additionally, an in-depth comparison was conducted between the most phylogenetically related carbapenemase-harboring plasmids and p4782-IMP using the Gview server’s BLAST Atlas. The analysis unveiled a distinctive transposon insertion within the MOB_F_/MPF_T_ family plasmid backbone, as illustrated in Figure 3. This novel transposon carried the novel *bla*_IMP-100_ allele, as well as multiple AMR determinants, which accounted for the atypical abundance of resistance genes observed in p4782-IMP.

### 3.6. Cloning, Transformation, and Conjugation

The transformants (PA01-p4782-IMP and *E. coli* NEB10-IMP-100) were successfully obtained. AST was conducted on both the recipients (PA01^RifR^ and *E. coli* NEB10) and the transformants. The results are provided in Table 2. Firstly, the introduction of the whole p4782-IMP into PA01 increased the MIC for imipenem at least eight-fold, and for meropenem at least 128-fold. In comparison, the IMP-100 gene alone in *E. coli* NEB10-IMP-100 increased the MIC for imipenem at least two-fold, and for meropenem at least sixteen-fold. Although in silico analysis with mob-suite suggested the transferability of the MDR plasmid, all mating experiments were unsuccessful for both *E. coli* NEB10 and PA01 despite numerous attempts and variations.

### 3.7. Phylogeny and Genetic Environment of blaIMP-100 in Paer4782MK

A phylogenetic analysis of all IMP variants was conducted through protein sequence alignment and subsequent construction of a phylogenetic tree (Figure 4). The clustering of IMP alleles and the labeling of the resulting groups were completed according to a clustering scheme suggested previously [14], thus maintaining consistency. Currently, there are 95 alleles including IMP-100. The novel allele IMP-100 falls into the G6 cluster revealing closest similarity to IMP-63, IMP-12, and IMP-90. The *bla*_IMP-100_ gene was positioned as the first cassette under a weak PcW+P2 promotor combination [64] in a novel *In1300*-like *Tn402*-type integron platform referred to as *In4886* and followed by *qnrVC4*, *cmlA5*, and *bla*_OXA-10_ cassettes (Figure 5).

A similar cassette array (*In1300*) was previously reported in the *Aeromonas salmonicida* MDR plasmid pS121-1a (CP022170). The *In4886* was preceded by a novel putative IS, hereby referred to as *IS4782* flanked by imperfect 25bp inverted repeats IRL (TGTCATTTTCAGAAG**G**CGACTGCAC) and IRR (TGTCATTTTCAGAAG**A**CGACTGCAC) closely resembling those of *ISPa17* and *Tn402.* The IRR was also found on the far 3′ end of the *Tn402* integron as well (Figure 4). The *IS4782* harbored three genes: the *yafQ* toxin gene, putative resolvase (188AA), and short transposase (128AA). This putative IS was lacking any significant homologs within the ISFinder or TnCentral databases [65,66]. The BLAST search was able to find eight identical hits (all plasmid-borne and one being the p3541_2 plasmid reported here) only matching the resolvase and transposase genes and not the whole IS. It was hypothesized that this element is capable of horizontal dissemination either alone or with the *Tn402* integron due to sharing the same inverted repeats. Next, we managed to identify and map related mobile elements in order to detect a hypothetical origin of the putative novel transposon, registered as *Tn7700* (Figure 5). The 5′ end most closely resembled the tnp/resolvase combination found in a *P. aeruginosa* transposon from the Czech Republic (KY860572) [67] followed by the *In4886* carrying IMP-100 that was highly similar to the one from the *Aeromonas salmonicida* MDR plasmid pS121-1a (Figure 5). The 3′ end of the transposon resembled a remnant from another one found in the *Enterobacter cloaceae* plasmid (AP022231) containing *katG*, *dps*, and *tniR*. Similar structures have been reported in most pMOS94-like plasmids where *ISPa17* preceded carbapenemase carrying *Tn402* integrons [63]. Recently it was hypothesized that the *ISPa17* transposase might be capable of the mobilization of adjacent *Tn402* integrons resulting in distinct transposon elements [68]. The *Tn7700* transposon had 5bp direct repeats (AAAAC) located up to 26bp apart from each IR, which although unexpected might be suggestive of past transposition events. Finally, we also discovered a probable ancestral insertion site of the whole *Tn7700* located in a *P. putida* plasmid from Brazil (CP016446) [8].

For the remaining two strains we found that IMP-13 (Paer3541) and IMP-84 (Paer3796A) were located chromosomally in class 1 integrons embedded in *Tn5051*-like transposons, whereas VIM-4 (in Paer4782MK) was found in the *In237*-like integron. Importantly, all these MBLs were components of self-transmissible T4SS-type ICEs meaning they are capable of dissemination by horizontal gene transfer (Appendix A. Lastly, all putative novel mobile genetic elements were submitted to the respective databases (ISFinder, TnRegistry, and INTEGRALL) [65,69,70].

## 4. Discussion

Between 2018 and 2022, three epidemiologically unrelated MDR *P. aeruginosa* strains were obtained from diverse clinical sources. Surprisingly, the modified CarbaNP test and the PCR for carbapenemase detection yielded conflicting results for strains Paer3541 and Paer3796A with the PCR being negative. Furthermore, WGS analysis revealed the presence of *bla*_IMP-13_ in Paer3541 and *bla*_IMP-84_ in Paer3796A. These findings were suggestive that the primers used to target *bla*_IMP_ genes were ineffective and had to be replaced. Paer4782MK was the most recent isolate of the three and the *bla*_IMP_ was therefore successfully detected with the updated PCR assay [27]. It is important to note that the availability of several alternative methods for carbapenemase detection may prove useful as well as regular updates of the methodologies being required to correctly detect the presence of rare gene variants.

Among the carbapenemases detected in Bulgaria, VIM-type enzymes are frequently observed [18,19], but occurrences of the NDM-1, GES-5, and OXA-50 carbapenemases have also been documented [20,21]. To our knowledge, this study represents the first report of *bla*_IMP_ detection in three clinical MDR isolates from Bulgaria. Additionally, we identified a novel allele of the *bla*_IMP_ gene, designated *bla*_IMP-100_, located on a MDR plasmid (p4782-IMP) within Paer4782MK. This strain also carried a chromosomally-encoded *bla*_VIM-4_ carbapenemase and was categorized as ST233, a globally prevalent multidrug-resistant clone [71], often associated with *bla*_VIM-2_ production [72]. Paer3541 (IMP-13) and Paer3796A (IMP-84) were classified as ST621, an epidemic clone known for its association with *bla*_IMP_ production and a higher prevalence of *pldA*, a trans-kingdom phospholipase T6SS effector [73]. This effector is associated with the H2 Type VI secretion system (H2-T6SS) and is involved in bacterial endocytosis [74].

In line with previous research demonstrating a correlation between ST233 and the *exoS*+ (*exoU*-) genotype [72], our virulence analysis revealed that Paer4782MK (ST233) exhibited this specific genotype. ExoU and ExoS are mutually exclusive T3SS effectors, with ExoS leading to delayed apoptotic cell death, while ExoU induces rapid host cell lysis [75]. Regarding serotypes, our findings align with existing literature [76], demonstrating a strong association between serotypes O4 and O11 and MDR phenotypes. Typically, the O4 serotype is associated with the *exoU*-negative genotype, while the O11 serotype is linked to the *exoU*-positive genotype. Interestingly, Paer4782MK tested negative for *exoU* and exhibited the *exoS* genotype. Furthermore, despite recent studies suggesting an association between ST233 and serotype O6 [77], our classification of Paer4782MK as ST233:O11 deviates from this anticipated relationship.

Upon analyzing the WGS data, it was determined that the observed negative expression of *oprD* in the expression assay was due to an identical 10 amino acid indel, which prevented primer binding across all three isolates, thereby rendering the results invalid. Additionally, the PorinPredict tool confirmed the presence of an intact and functional *oprD* porin in Paer3541, which likely contributed to its susceptibility to and the lowest MIC values of meropenem, imipenem, meropenem/vaborbactam, and imipenem/relebactam for all three strains. Despite the occurrence of a missense mutation S325F, the *oprD* integrity in Paer3796A remained unaffected. In fact, the presence of this mutation has been associated with a potential increase in the MIC of imipenem [78]. However, further investigations are necessary to determine its precise impact. Lastly, both our observation and PorinPredict concluded that the *oprD* gene in Paer4782MK is truncated, and therefore its functionality is compromised, consistent with the higher reported MIC values for meropenem, imipenem, meropenem/vaborbactam, and imipenem/relebactam in this particular strain.

Phylogenetic analysis demonstrated a close genetic association between the plasmid p4782-IMP, harboring the novel *bla*_IMP-100_ allele, and the pMOS94-like plasmid family. The pMOS94 plasmid family was recently recognized as an emerging lineage involved in the dissemination of MBL genes among *Pseudomonas* species, also commonly carrying *bla*_VIM_ and *bla*_IMP_ genes, and recently *bla*_BIM_, and some of them were found to display disruptions in the transfer module (*trw*), thus impeding conjugation [63]. In the genetic environment of the *trw* transfer module of p4782-IMP, we could not identify any disruption, so the observed unsuccessful mating experiments could be due to other reasons. Next, the comparison with the nearest carbapenemase-harboring plasmids revealed an insertion of the unique *Tn7700* transposon in p4782-IMP, carrying multiple AMR determinants. The transformation of PA01 with the p4782-IMP plasmid resulted in a resistance profile similar to that of the Paer4782MK strain, demonstrating the impact of p4782-IMP in reducing susceptibility to important antibiotic classes and conferring multiple drug resistance. Furthermore, the transfer of the entire plasmid into the PA01 conferred resistance to CFDC. On the other hand, in the case of the *E. coli* NEB10 transformant carrying only the *bla*_IMP-100_ gene, there was no significant increase in the CFDC MIC suggesting that the CFDC resistance was a result of the complex interplay between multiple genes present on the plasmid, in addition to *bla*_IMP-100_.

## 5. Conclusions

Our study is the first to our knowledge to document the *bla*_IMP_ in clinical *P. aeruginosa* from Bulgaria. We also discovered a novel *bla*_IMP-100_ allele located on a MDR plasmid (p4782-IMP). The plasmid in turn conferred resistance to multiple antibiotics including CFDC, as shown by the similar MDR pattern revealed in the PA01 transformant. However, by examining transformants carrying only the *bla*_IMP-100_ gene, we observed that the CFDC MIC values did not show a substantial increase, further implying the involvement of other plasmid genes in the development of resistance to this novel agent. Lastly, our study emphasizes the importance of employing multiple methods for carbapenemase screening, given that conventional methods, apart from WGS, might show reduced sensitivity and neglect carbapenemase activity or genes.

## Figures and Tables

**Figure 1 microorganisms-11-02270-f001:**
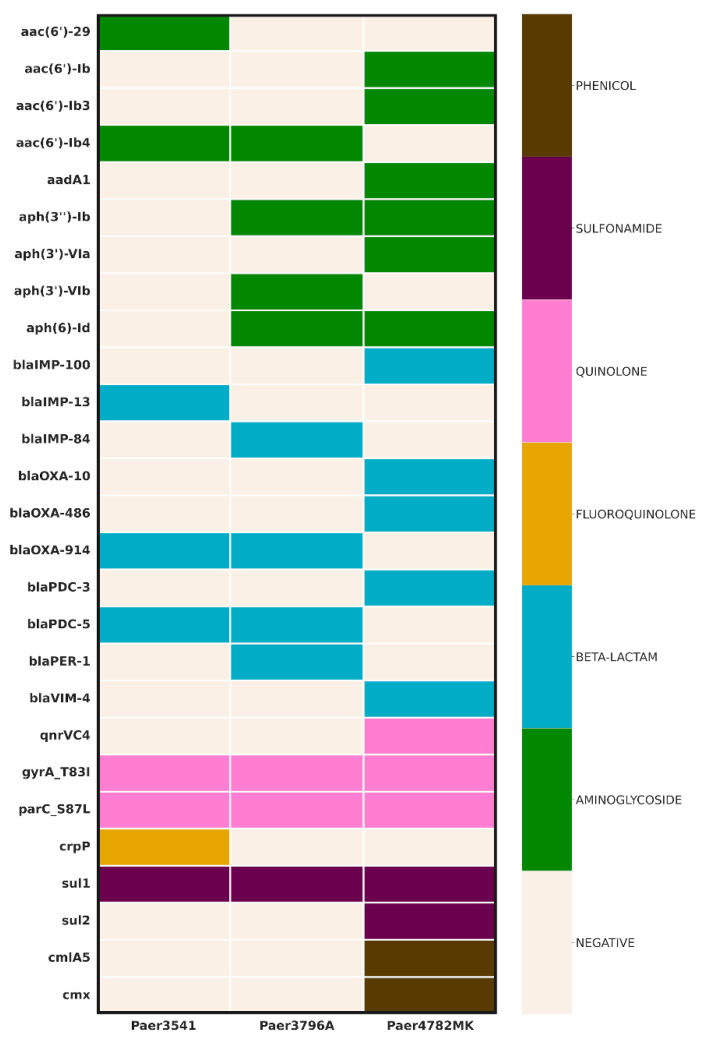
Genomic inference of resistance determinants. Columns represent strains, whereas AMR genes are in rows. The color scheme represents the affected antimicrobial class (**right**) in relation to the resistance genes (**left**).

**Figure 2 microorganisms-11-02270-f002:**
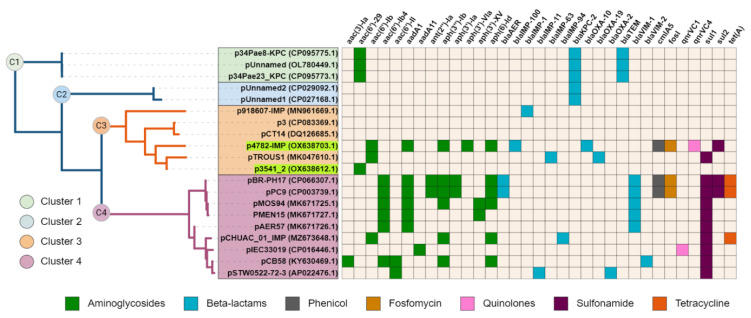
Phylogenetic tree and resistance heatmap. Plasmids from the current study are highlighted in green. The different tree clusters (C1 ÷ C4) are distinguished by varying colors. The heatmap legend below provides information on the antibiotic class of the detected genes. The tree was generated with neighbor-joining after the NCBI blast with Max Seq Difference score ≥0.75. The heatmap and the tree were visualized with iToL 6.8 (https://itol.embl.de/about.cgi, accessed on 25 July 2023).

**Figure 3 microorganisms-11-02270-f003:**
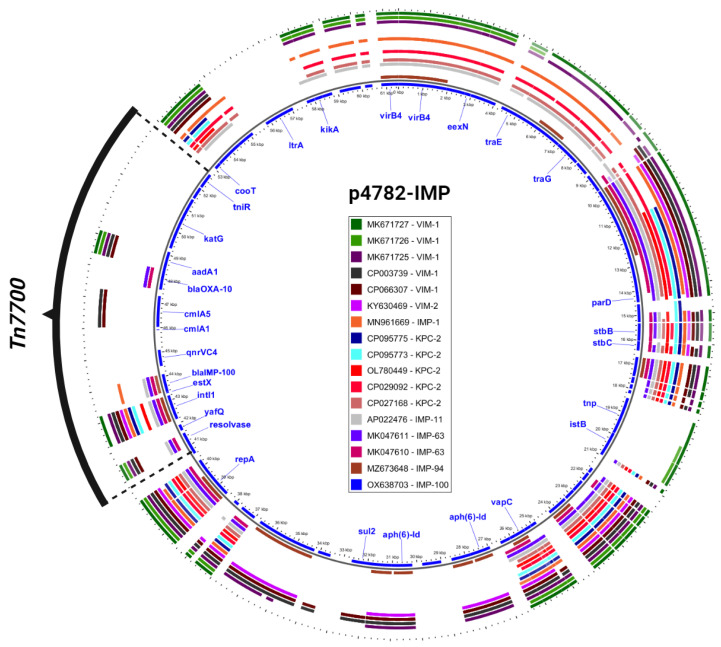
Closest plasmids with carbapenemases. The plasmid accession numbers are listed in the legend and colored differently. The most inner circle (blue) is p4782-IMP which serves as a reference for comparison. Only important gene annotations were visualized in blue text. The novel transposon (Tn7700) is shown in black. The figure was produced with the Gview Atlas server (https://server.gview.ca, accessed on 25 July 2023).

**Figure 4 microorganisms-11-02270-f004:**
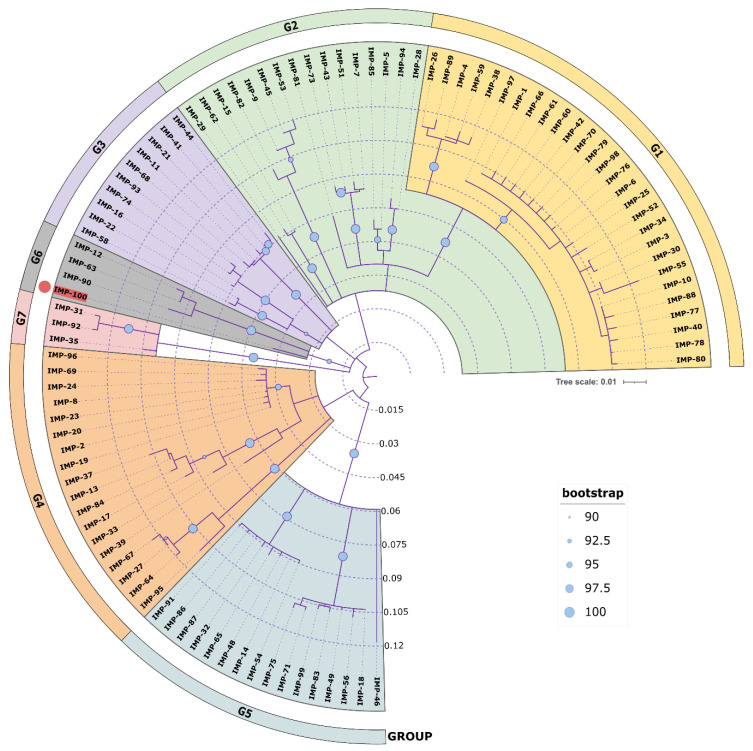
Phylogenetic tree of all available IMP variants. All NCBI available protein sequences were downloaded and aligned with *Muscle* alignment 3.8.425 in Geneious Prime 2022 and the phylogenetic tree was constructed with Geneious Tree builder with the Jukes–Cantor genetic distance model and the neighbor-joining build method with bootstrap 1000. The tree was visualized in iToL 6.8 (https://itol.embl.de/about.cgi, accessed on 25 July 2023). Cluster groups G1 ÷ G7 were colored differently in accordance to [14], and the new IMP-100 allele is colored in red with a red dot at the tip of the leaf. Bootstrap indices are represented as circles placed at each node ranging in size from 90 (small circle) to 100 (large circle) to show the degree of cluster consistency.

**Figure 5 microorganisms-11-02270-f005:**
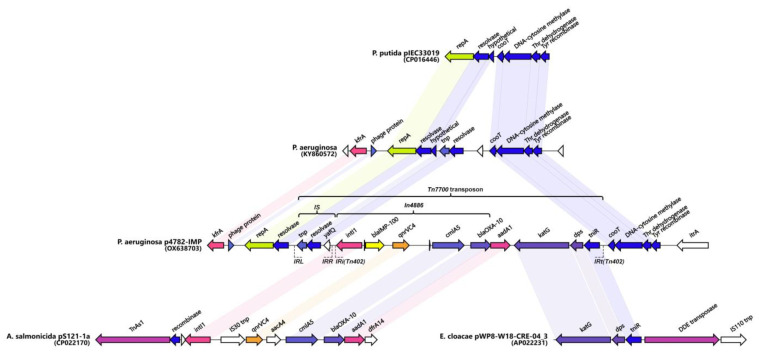
Genetic environment of *bla*_IMP-100_. A hypothesized origin of the novel transposon *Tn7700* is depicted. Matching genes from the different sources are connected and identically colored. Non-matching genes (either hypothetical proteins or not) are shown in white. Unlabeled genes are hypothetical. The figure was created with Clinker v0.0.28 (https://github.com/gamcil/clinker, accessed on 25 July 2023).

**Table 1 microorganisms-11-02270-t001:** Minimal inhibitory concentrations (MIC) of *P. aeruginosa* strains in this study.

	MIC (μg/mL)					
Antimicrobials	Paer4782	SIR *	Paer3541	SIR	Paer3796A	SIR
Piperacillin	≥32	R	16	I	8	I
Piperacillin/Tazobactam	64/4	R	16/4	I	8/4	I
Cefepime	≥8	R	≥8	R	≥8	R
Ceftazidime	≥32	R	≥32	R	≥32	R
Ceftazidime/Avibactam	≥8/4	R	≥8/4	R	≥8/4	R
Ceftolozane/Tazobactam	≥8/4	R	≥8/4	R	≥8/4	R
Cefiderocol	4	R	0.25	S	8	R
Imipenem	≥8	R	1	I	4	I
Imipenem/Relebactam **	≥32/4	R	0.75/4	S	1.5/4	S
Meropenem	≥16	R	0.5	S	≥16	R
Meropenem/Vaborbactam **	≥256/8	R	0.5/8	S	≥256/8	R
Doripenem **	≥32	R	2	I	≥32	R
Aztreonam	4	I	8	I	≥16	R
Trimethoprim/Sulfamethoxazole	≥8/152	n/a	≥8/152	n/a	≥8/152	n/a
Amikacin	≥32	R	16	S	4	S
Tobramycin	≥32	R	16	R	8	R
Gentamicin	16	n/a	≥32	n/a	≥32	n/a
Ciprofloxacin	≥8	R	≥8	R	2	R
Levofloxacin	≥8	R	≥8	R	2	I
Fosfomycin	32	n/a	≥128	n/a	<16	n/a
Colistin	<1	S	<1	S	<1	S

* SIR—Susceptible (S), susceptible to increased exposure (I), resistant (R); **—gradient strip; n/a—not applicable.

**Table 2 microorganisms-11-02270-t002:** AST of *P. aeruginosa* and *E. coli* transformants.

	MIC (μg/mL) for Strain				
	Donor Strain	*Pseudomonas aeruginosa*		*Escherichia coli*	
Antimicrobials	Paer4782	PA01	PA01-p4782-IMP	*E. coli* NEB10	*E. coli* NEB10-IMP-100
Piperacillin	≥32	<4	≥32	<4	4
Piperacillin/Tazobactam	64/4	1/4	64/4	<1/4	4/4
Cefepime	≥8	<1	≥8	<1	≥8
Ceftazidime	≥32	0.5	≥32	<0.25	≥32
Ceftazidime/Avibactam	≥8/4	<1/4	≥8/4	<1/4	≥8/4
Ceftolozane/Tazobactam	≥8/4	<1/4	≥8/4	<1/4	≥8/4
Cefiderocol	4	0.25	8	0.125	0.125
Imipenem	≥8	<1	≥8	<1	2
Imipenem/Relebactam *	≥32/4	0.25/4	8/4	0.125/4	2/4
Meropenem	≥16	<0.125	≥16	<0.125	2
Meropenem/Vaborbactam *	≥256/8	0.25/8	48/8	≤0.016/8	2/8
Doripenem *	≥32	0.1	≥32	0.08	6
Ertapenem *	n/a	n/a	n/a	0.023	6
Aztreonam	4	1	8	<1	<1
Trimethoprim/Sulfamethoxazole	≥8/152	1/19	1/19	<1/19	<1/19
Amikacin	≥32	<4	4	<4	<4
Tobramycin	≥32	<0.25	8	<0.25	0.5
Gentamicin	16	0.25	16	0.25	0.25
Ciprofloxacin	≥8	0.0625	1	<0.06	0.06
Levofloxacin	≥8	0.125	1	<0.125	<0.125
Fosfomycin	32	16	≥128	<16	<16
Colistin	<1	<1	<1	<1	<1

*—gradient strip; n/a—not applicable.

## Data Availability

All used data are included in the main text and in the Appendix A. Relevant links and/or references to other sources are included in the main text. The generated information and/or datasets analyzed during the current study are available from the corresponding author on reasonable request.

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
