# Peer review of "Genomic Characterization of IMP-Producing Pseudomonas aeruginosa in Bulgaria Reveals the Emergence of IMP-100, a Novel Plasmid-Mediated Variant Coexisting with a Chromosomal VIM-4â€"

_microorganisms, 2023, doi:10.3390/microorganisms11092270_

Round 1
Reviewer 1 Report
The article provided by the authors provide a rigorous investigation of the genetic landscape of P. aeruginosa in Bulgaria, which is useful for both local and world-wide epidemiology.
I would ask if the authors have used immunochromatographic tests (which are becoming more and more useful for both hospital and reference center) for the identification of the VIM and IMP carbapenemases, as this may provide valuable information in absence of the alternative primer sets that the authors have used.
The use of English in this article is excellent, comprehensive, easy to read.
I found only one obvious flaw in the abstract, where it is written „..infections represent major a public health concern..” - probably an inversion of „.. represent a major public health..”
Reviewer 2 Report
This study investigated the first occurrence of IMP-carrying MDR P. aeruginosa strains from Bulgaria. Whole genome sequencing identified a novel plasmid-mediated blaIMP-100 allele located in a In1300- like integron embedded in a novel putative transposon. However, the data in the manuscript did not support the conclusion solidatelly. Here are some concerns.
1. The introduction is confusing due to using lots of abbreviations without full names and the lack of logical organization. The background of MBLs should be illustrated a little bit, such as their different types, and their encoding genes, particularly IMP and VIM, their role in antibiotic resistance, and the classification of IMP.
2. What does BlaIMP represents? The full name should be provided.
3. The question and significance of the study, particularly the identification of IMP-100, should be explicitly stated in the introduction.
4. The small sample size of three P. aeruginosa strains analyzed should be justified. Explain why these three strains were selected or if they hold any special characteristics.
5. Please provide detailed methods for the biofilm formation assay.
6.In the methods, please provide how the fragment of bla IMP-100 was obtained?
7. What do WGS and NGS represent?
8. Provide a brief explanation of how in-silico serotyping is performed.
9 In result 1, “The AST profiles of the three IMP-carrying P. aeruginosa isolates”. How these 3 isolates are defined as IMP-carring?
10. In Table 1, SIR, MIC la should be noted with the full names.
11. The novel blaIMP-100 was originally identified in this study? If so, Highlight that the novel blaIMP-100 was identified for the first time in this study. Describe how this plasmid was identified, named, and emphasize its novelty.
12.Please provide some background information on MOB and KPC for better understanding of the results.
13. Page 7, Paer3541 and Paer3796A were assigned to ST621. Why ST was determined? What is the relationship of this with the novel plasmid? Please provided brief rationale of determining ST.
14. Clarify the color scheme and column meanings in Figure 1.
15. In the Results section "Detection of Virulence Determinants," the description of the results is kind of confusion. Please clearly mention which isolates express which genes and which ones do not.
16. In the result “Plasmid analysis”, p3782-IMP was not list in the figure 2.
17. Explain why plasmid p3782-IMP, despite having a high abundance of AMR genes similar to C4 plasmids, was grouped within the lower-resistance cluster C3.
17. Despite the high abundance of AMR genes found on plasmid p4782-IMP similar to C4 plasmids, it was grouped within the lower-resistance cluster C3. Please discuss why?
18. In the Results section "Phylogeny and Genetic Environment of blaIMP-100 in Paer4782MK," provide background information on IMP variant classification and the current number of variants.
19.Throughout the manuscript, ensure all abbreviations are provided with their full names. Examples include IRL, IRR, ESKAPE, OXA, GES, VIM, SPM, GIM, NDM, FIM, BIM, NDM-1, GES-5, OXA-50, WGS, ESBL, PER-1.
